# Exploring the Potential Role of Circulating microRNAs as Biomarkers for Predicting Clinical Response to Neoadjuvant Therapy in Breast Cancer

**DOI:** 10.3390/ijms24129984

**Published:** 2023-06-10

**Authors:** Luis M. Ruiz-Manriquez, Cynthia Villarreal-Garza, Javier A. Benavides-Aguilar, Andrea Torres-Copado, José Isidoro-Sánchez, Carolina Estrada-Meza, María Goretti Arvizu-Espinosa, Sujay Paul, Raquel Cuevas-Diaz Duran

**Affiliations:** 1School of Medicine and Health Sciences, Tecnologico de Monterrey, Monterrey 64700, Mexico; a01701195@tec.mx; 2School of Engineering and Sciences, Tecnologico de Monterrey, Queretaro 76130, Mexico; 3Breast Cancer Center, Hospital Zambrano Hellion TecSalud, Tecnologico de Monterrey, Monterrey 64700, Mexico; cynthia.villarreal@tecsalud.mx

**Keywords:** miRNA, circulating miRNAs, breast cancer, neoadjuvant therapy

## Abstract

Breast cancer (BC) is a leading cause of cancer-related deaths among women worldwide. Neoadjuvant therapy (NAT) is increasingly being used to reduce tumor burden prior to surgical resection. However, current techniques for assessing tumor response have significant limitations. Additionally, drug resistance is commonly observed, raising a need to identify biomarkers that can predict treatment sensitivity and survival outcomes. Circulating microRNAs (miRNAs) are small non-coding RNAs that regulate gene expression and have been shown to play a significant role in cancer progression as tumor inducers or suppressors. The expression of circulating miRNAs has been found to be significantly altered in breast cancer patients. Moreover, recent studies have suggested that circulating miRNAs can serve as non-invasive biomarkers for predicting response to NAT. Therefore, this review provides a brief overview of recent studies that have demonstrated the potential of circulating miRNAs as biomarkers for predicting the clinical response to NAT in BC patients. The findings of this review will strengthen future research on developing miRNA-based biomarkers and their translation into medical practice, which could significantly improve the clinical management of BC patients undergoing NAT.

## 1. Introduction

According to the World Health Organization, BC is the most frequent cancer diagnosed in women worldwide, with an estimated 2.3 million new cases and 685,000 deaths in 2020 [1]. BC is biologically and clinically diverse, with several recognized histotypes and molecular subtypes with various etiologies, risk factor profiles, prognoses, and therapeutic response rates [2]. The optimal therapy is multimodal, and the treatment sequence considers molecular subtypes and locoregional tumor burden [3]. NAT refers to the systemic treatment of BC before definitive surgical therapy (i.e., preoperative therapy) [4]. The mechanisms of action of NAT through chemotherapy agents, targeted therapy, and hormone therapy have been thoroughly revised [5,6,7]. Initially, NAT was intended for patients with locally advanced or inoperable BC, with the main goal of reducing the tumor (also known as downstaging) to enable breast conservation surgery. However, the role of NAT has expanded to include patients with early stage, operable BC [8]. Furthermore, in recent years, there has been a growing interest in determining the role of neoadjuvant endocrine therapy and targeted therapies, either in conjunction with chemotherapy or as a monotherapy [9,10].

Pathologic complete response (pCR) (defined as the absence of residual disease or eradication of invasive disease following NAT) has been considered a potential surrogate marker for disease-free and overall survival [11]. Therefore, improving pCR rates has become the goal of NAT with the expectation of enhancing oncological outcomes [12]. However, pCR rates differ between the diverse BC subgroups: 0–8% for luminal A, 15% for luminal B HER2-, 22–48% for luminal B HER2+ with trastuzumab combined in NAT, and more than 50% for HER2-enriched and triple-negative BC (TNBC) [12,13,14]. Despite being hypothesized that overall survival might be improved with NAT, randomized trials have confirmed equivalent mortality for the pre- or postoperative delivery of similar systemic therapy [15]. Nevertheless, the NAT provides a potentially efficient trial design to explore the efficacy of novel therapies [16]. For example, researchers collect imaging data, tumor specimens, and blood samples before and throughout NAT, during surgery (in patients with sufficient residual disease), and after preoperative treatment [8]. This sampling strategy has proven helpful in identifying tumor- or patient-specific biomarkers of response or resistance [17]. Undoubtedly, such biomarkers could help reduce chemotherapy-related toxicities and the potential risk of distant metastasis and facilitate tailored clinical management [18]. Therefore, the early recognition of a non-responsive patient facilitates an early change to a non-cross-resistant regimen, thereby minimizing toxicity and optimizing the timing of the surgery [19].

As the achievement of pCR translates into a favorable prognosis, research increasingly concentrates on identifying novel biomarkers to monitor response to the NAT regimen, thus adapting the treatment to individualized patient risk [20]. Recently, non-invasive biomarkers found in blood (i.e., liquid biopsies) have been proposed as a feasible method to discern BC subtypes and predict response to therapy, since it provides a dynamic panorama of the entire tumor burden at specific time points. This allows the identification of changes reflecting an individual’s response to treatment [21]. Over the past few years, a plethora of evidence has highlighted the potential use of peripheral blood circulating nucleic acid molecules in BC diagnosis, prognosis, and monitoring response to therapy [22,23,24]. Among these, circulating miRNAs are increasingly recognized as promising biomarkers, given their stability, non-invasive, real-time, and easily accessible sampling characteristics [25].

miRNAs are short single-stranded non-coding RNA molecules (20–24 nucleotides) that mediate posttranscriptional gene regulation [26,27]. miRNAs induce decay and the translational suppression of mRNAs by binding to coding regions or interacting with complementary sequences in the 3′-untranslated regions (3′-UTR) [28]. Hitherto, the high expression of miRNAs may synergistically reduce the concentration of specific mRNAs. Inversely, low miRNA levels are associated with higher mRNA concentration [29]. Nearly 60% of the human genome is regulated by miRNAs, suggesting that they are involved in almost all essential cellular functions [30,31]. Mounting evidence has demonstrated that miRNA expression is frequently altered in complex pathophysiologies, including cancer [32,33,34]. Many studies have also confirmed the crucial role of miRNAs in various cancer-associated biological processes, such as proliferation, differentiation, apoptosis, metabolism, invasion, metastasis, and drug resistance in almost all cancer types, including BC [35,36,37,38]. The aberrant expression of miRNAs in cancer patients can be attributed to various mechanisms [39]. Nearly half of the genes coding for miRNAs are located in cancer-associated loci, where they are translocated or activated during carcinogenesis [40]. Alterations in the activities of the enzymes responsible for miRNA biogenesis, such as Drosha and Dicer, can also result in abnormal miRNA levels [41]. In cancer, variations in circulating miRNAs may be induced by pri-miRNA (intermediate product in miRNA biogenesis) transcriptional errors [42]. These factors contribute to the distinctive miRNA profiles observed in different types of cancer, making them potential biomarkers for early diagnosis, prognosis, or predicting therapeutic outcomes [43].

Extracellular miRNA-dependent cell–cell communication has been investigated since miRNAs exist in a stable cell-free form in body fluids and other extracellular environments, including plasma, serum, urine, and saliva [44]. These molecules are released encapsulated in cell-free lipid carriers (microvesicles, exosomes, and apoptotic bodies) or bound to protein complexes with Argonaute or high-density lipoproteins (HDL). This mechanism enables miRNAs to evade RNase digestion and remain stable in circulation [45], thus positioning them as attractive potential biomarkers for tracking disease development and progression [39,46]. The dysregulation of circulating miRNA levels may be a result of tumor cell secretion, the altered activity of the heterogeneous collection of cells present in the tumor microenvironment or from tumor cell death lysates. However, the exact mechanisms underlying this alteration remain elusive. It has been demonstrated that primary tumor-derived miRNAs can be transferred to nonmalignant cells in the tumor microenvironment during tumor progression to induce heterogeneity [47,48,49]. Simultaneously, non-malignant cells can also secrete miRNAs to regulate tumor progression or other microenvironmental components [47,50]. Therefore, although many circulating miRNAs in cancer patients may precisely not originate from tumor cells, their expression directly reflects the body’s homeostatic response to the disease [47,51]. Consequently, circulating miRNA profiles are promising biomarkers for early diagnosis, treatment sensitivity/resistance, and prognosis in different cancer types, including BC (Figure 1) [25]. Herein, we will review the potential predictive value of circulating miRNAs regarding treatment response in BC patients receiving a NAT (Table 1).

## 2. Circulating miRNAs as Indicators of Clinical Response to NAT in BC

Over the past decade, mounting evidence has indicated that the differential expression profile panels of circulating miRNAs may be useful in stratifying patients who are more likely to respond positively to NAT. For example, in a prospective cohort, W. Zhu et al. [36] investigated how changes in plasma miRNA levels correlate to disease response during NAT. The authors collected peripheral blood samples at three-time points: baseline, after two cycles of chemotherapy (C2), and before surgery, and the plasma levels of miRNAs were assessed to examine their dynamics throughout the treatment. Interestingly, three plasma miRNAs (miR-222, miR-20a, and miR-451) were identified as being able to predict chemosensitivity in the HR+/HER2- cohort. Specifically, a high level of miR-222 (*p* = 0.049) in the plasma at the baseline was linked to a poor response to NAT. Additionally, chemotherapy led to a further upregulation of miR-222 in patients who were insensitive to the treatment. The authors suggested that miR-222 might be involved in the mechanism responsible for the resistance to anthracyclines/taxanes. Notably, other studies demonstrated that miR-222 hinders the expression of trichorhinophalangeal syndrome type 1 protein (TRPS1) and Notch-3, while simultaneously promoting epithelial-to-mesenchymal transition (EMT) [67,68], which ultimately leads to a chemo-resistance and aggressive BC phenotype. Conversely, the baseline levels of circulating miR-20a and miR-451 did not correlate with chemosensitivity, but their dynamics (measured by C2 fold change) predicted the ultimate clinical response. Specifically, the authors proposed that C2 miR-451 (*p* = 0.012) downregulation and miR-20a upregulation (*p* = 0.021) may be early markers of chemosensitivity in HR+/HER2− BC. Intriguingly, miR-451 regulates the expression of multidrug resistance protein 1 (MRP-1), a key player in modulating the insensitivity to anthracyclines [69,70], while miR-20a expression is regulated by the well-known oncogenic factor c-Myc, which is closely involved in tumorigenesis in BC [71]. Moreover, it has been shown that miR-20a inhibits the expression of tumor suppressor ZBTB4 protein, which is also related to the suppression of the growth and invasion of BC cells [72]. Furthermore, the authors evaluated the correlation between decreased expression of C2 miR-34a and response to NAT across all BC subtypes. Their findings indicated that plasma miR-34a has the potential to serve as a biomarker of clinical response to NAT regardless of BC subtype. Notably, the transcription factor p53 directly regulates miR-34a expression by binding to its promoter, which in turn promotes cellular apoptosis by targeting Bcl-2 and SIRT1 [73]. These serial assays of circulating miRNAs during NAT provided valuable insights into the significant alteration patterns in plasma miRNAs and their potential as a novel response indicator, which could guide the personalized delivery of NAT.

The early identification of non-responding BC patients in neoadjuvant settings is crucial for optimizing treatment strategies and improving patient outcomes, particularly for aggressive TNBC and HER2+ BC subtypes, since achieving pCR in these subtypes has been substantially associated with improved long-term outcomes [74,75]. A study by Stevic et al. [53] assessed a specific miRNA signature in exosomes that could predict pCR to NAT. Using plasma samples from BC (HER2+ and TNBC) patients after NAT before surgery, the authors showed that a downregulation of exosomal miR-155 (in both TNBC and HER2+BC subtypes) (*p* = 0.002) and miR-301 (in HER2+BC patients) (*p* = 0.002) significantly predicted pCR to NAT. Strikingly, miR-155, a well-known key modulator in BC carcinogenesis, has been found to downregulate ErbB2 by directly or indirectly targeting HDAC2, a transcriptional activator of ErbB2 in BC, closely regulating cell survival, growth, and chemosensitivity [76]. Moreover, it has been previously reported that miR-301 directly binds to estrogen receptor 1 (ESR1) mRNA, which finely modulates estrogen signaling pathways that are crucial in the progression of BC [77,78]. Together, these findings suggest that specific miRNAs are selectively enriched in exosomes of HER2+ and TNBC patients and are also associated with achieving pCR, providing an insight into exosome biology for monitoring the disease and positioning exosomal miRNAs as potential diagnostic markers and therapeutic molecules. 

Rodriguez-Martínez et al. [62] analyzed peripheral blood samples from localized BC women. Samples were extracted at diagnosis time and after four cycles of doxorubicin/cyclophosphamide, from which they isolated circulating tumor cells (CTCs) and exosomes. Results revealed that patients with higher serum miR-21 levels (*p* = 0.039) tended to have larger tumors; however, after four cycles of doxorubicin/cyclophosphamide, lower levels of miR-21 were observed in HER2 + patients during NAT with trastuzumab. In addition, an inverse association (*p* = 0.031) was observed between miR-21 and Ki67 expression after four treatment cycles [62]. They also reported that exosomal miRNA-21 and miRNA-105 expression levels were higher in metastatic patients than in non-metastatic and healthy subjects. Moreover, exosomal miRNA-222 correlated with variables such as progesterone receptor (PGR) status (*p* = 0.017) and Ki67 expression (*p* = 0.05). A correlation between CTCs at baseline status and higher levels of miR-21 (*p* = 0.032), miR-155 (*p* = 0.039), and miR-222 (*p* = 0.019) miRNAs was also shown, which has been associated with tumor aggressiveness potentially by promoting the proliferation and migration of tumor cells. According to Rodriguez-Martínez et al. [62], low miR-21 expression might be caused by MAPK (ERK1/2) pathway blockage. Furthermore, miR155-5p was found to be closely related to the status of the estrogen receptor (ER) and PGR [79]. Ali et al. [80] revealed that miR-155 targets PTEN, a crucial tumor suppressor in BC biopsies. Finally, miR105 was shown to induce migration via the downregulation of ZO-1 in BC cells [81], and miR-222 and miR221 directly target ERα transcripts, conferring BC cells with a proliferative advantage and migratory activity, thus promoting the transition from ER+ to ER− tumors [82]. These results suggest that exosomal miRNAs from liquid biopsies may serve as novel non-invasive biomarkers in clinical practice to improve BC diagnosis [62].

In 2019, Di Cosimo et al. [56] studied plasma miRNA levels in HER2+BC patients to predict the therapeutic response to NAT. In this study, phase III HER+BC patients were randomly treated with lapatinib, trastuzumab, or their combination for 6 weeks, where the main endpoint was pCR followed by event-free survival (EFS). Blood samples were obtained at baseline (T0) and after 2 weeks of treatment (T1) to identify early pCR predictors. Using a *p* value < 0.05, the authors defined four miRNA signatures associated with pCR in the training set. For instance, with lapatinib at T0, miR-376c-3p and miR-197-3p were downregulated, whereas miR-874-3p, miR-320c, and miR-100-5p were overexpressed. In the same treatment, at T1, miR-144-3p, miR-362-3p, and miR-100-5p were highly expressed. For trastuzumab at T1, miR-374a-5p, miR-574-3p, miR-140-5p, and miR-145-5p were substantially upregulated, while miR-328-3p was downregulated. Finally, with lapatinib + trastuzumab at T1, miR-34a-5p, and miR-100-5p were downregulated and miR-98-5p was overexpressed. Among the aforementioned miRNA signatures, only lapitanib at T0 was not proven to be time- nor treatment-specific. Moreover, the authors evaluated whether the verified miRNA signatures were related to EFS; nevertheless, only miR-140-5p of the trastuzumab T1 signature was found to be significantly associated with EFS. The majority of the miRNAs found to be enriched in the aforementioned scenarios have been demonstrated to regulate genes related to BC. For example, miR-376c-3p targets RAB2A to regulate the properties and fates of BC stem cells (BCSC) [83], and miR-197-3p targets HIPK3, which is associated with cell colony formation, progression, migration, and apoptosis in BC cells [84]. Moreover, miR-874-3p promotes TNBC progression via SOX2, whereas miR-320 regulates AQP1 and the PI3K/AKT signaling pathway, inhibiting tumor progression in BC [85,86]. Fuso et al. [87] reported that miR-100-5p modulates the PlK1 gene and Wnt/β-catenin pathway, acting as a pro-differentiating agent in BCSC. MiR-144-3p and miR-145-5p target SOX2 in BC cells and are involved in cell migration, invasion, stemness, and cancer progression [88,89]. The hERG gene, which is related to cell proliferation, differentiation, and apoptosis in several types of cancers, including BC, is targeted by miR-362-3p [90]. Son et al. [91] found that miR-374a-5p promotes tumor progression by directly targeting ARRB1 in TNBC, whereas in liver cancer, miR-574-3p targets ADAM28, a gene related to tumorigenesis. However, this gene has also been observed to be overexpressed in BC [92]. miR-140-5p inhibits proliferation of BCSC and increases doxorubicin efficacy by targeting Wnt1, a crucial regulator of the Wnt/β-catenin pathway [93]. Recently, Deng et al. [94] noticed that miR-34a could inhibit tumor growth, cell activity, and migration and can promote cytotoxicity and apoptosis in TNBC through PD-L1 and p53; and miR-98-5p regulates cell proliferation, invasion, and migration by targeting IGF1 in BC tissues [95]. Hence, differential miRNA expression during BC treatment can provide a novel tool for better understanding how anticancer drugs act alone and in combination, allowing treatment optimization [56].

In 2019, B. Liu et al. [58] evaluated the expression of serum miRNAs at different time points during treatment with NAT, including trastuzumab, in HER2+BC patients. It was observed that at baseline, miR-210 was underexpressed (*p* < 0.001), while miR-10b, miR-21, miR-34a, miR-125b, miR-145, miR-144, and miR-373 were overexpressed (*p* < 0.001 for all seven miRNAs) in healthy controls than in HER2+ patients. Moreover, the overexpression of serum miRNAs was correlated with several clinicopathological parameters. For example, the overexpression of miR-10b, miR-125b, and miR-373 was associated with lymph node status, whereas the upregulation of miR-145 and miR-10b was associated with ER positivity and negativity, respectively. Among the miRNAs, only miR-21 levels were significantly decreased after therapy (*p* < 0.001). The upregulation of miR-21 has been associated with the cell proliferation, self-renewal, and induction of EMT in cancerous cell lines and tissues [58]. Previously, miR-21 was identified as an oncogenic miRNA, with PTEN being one of its more relevant targets responsible for the induction of apoptosis and the inhibition of cell proliferation [96]; hence, the lower expression of serum miR-21 after neoadjuvant chemotherapy combined with trastuzumab signifies the positive effect of the therapy in the patient by potentially increasing PTEN titers, thereby inducing the apoptosis of cancerous cells.

It has been demonstrated that a higher number of CTCs increases the risk of BC metastasis and reduces the survival rates of patients [97]. Akkiprik et al. [54] investigated the potential of differential expression profiles of miRNAs and CTCs in predicting the response and early recurrence of locally advanced BC in blood plasma samples extracted before and after NAT. Interestingly, authors found a six-fold decrease (*p* = 0.008) in miR-146b-5p expression in CTC+ patients compared to CTC− patients before treatment. Moreover, post-treatment CTC+ patients showed a five-fold increase (*p* = 0.014) in miR-199a-5p expression compared to post-treatment CTC− patients. Notably, both miRNAs were shown to suppress proliferation, migration, and stem cell-like characteristics by targeting EMT-related genes, such as the EMT inducer ZEB1 (miR-146b-5p) [98], CDH1 and TWIST (miR-199a-5p) [99]. These results suggest that miRNA expression profiles are linked to CTC status and may be critical in determining metastasis progression after treatment and useful in monitoring therapy response.

Ibrahim et al. [61] collected three blood samples from patients with BC at three different times: baseline before the beginning of NAT, after four cycles of adriamycin/cyclophosphamide (inter-treatment), and after 12 doses of weekly paclitaxel. Plasma expression of miRNAs was evaluated. The results showed that miR-10b, miR-21, and miR-181a significantly increased in patients with locally advanced BC (*p* < 0.01, *p* < 0.001, and *p* < 0.05, respectively), whereas miR-145 and let-7a were significantly downregulated (*p* < 0.01 both) compared to healthy subjects. The expression levels of plasma miR-21 were sufficient to discriminate locally advanced tumor subtypes into luminal A, luminal B, HER2, or TNBC. Additionally, increased levels of miR-10b and miR-21 were found in patients with locally advanced BC suffering from relapse or metastases to lungs, liver, or bones during a 5-year follow-up after tumor resection [61]. Thus, these miRNAs are potentially good predictors of relapse, metastasis, and even recovery. Interestingly, the expression of miR-10b, miR-21, miR-145, miR-181a, and let-7a is closely associated with the clinical and pathological features of BC, for example lymph node status, tumor size, and expression of sex hormones [61]. MAT2, STARD13, and ZNF132, targets of miR-21-3p, have a role in migration and metastasis in BC [100], and miR-145-5p plays a suppressive role in the proliferation of BC cells by targeting SOX2 [89]. In a similar study, Jiang et al. [101] reported that HBXIP, an oncogenic co-activator, was diminished by tumor-suppressive miR-145 in BC cultures. Another study on TNBC cells [102] found that miR-10b-5p regulated five target genes involved in sustaining proliferative signaling in cancer development: BIRC5, E2F2, KIF2C, FOXM1, and MCM5. Similarly, Zhai et al. [103] analyzed tumor tissues and noticed that the tumor suppressor NDRG2 is a target of miR-181a-5p, promoting proliferation and migration through the PTEN/AKT pathway. Hence, miR-10b, miR-21, miR-181a, miR-145, and let-7a are potentially non-invasive diagnostic biomarkers of locally advanced BC and miR-10b and miR-21 may be considered predictive biomarkers for progression-free survival [61].

Chekhun et al. [64] associated serum miRNAs with NAT response in stage II-III BC patients with luminal A and B subtypes. They analyzed the levels of circulating miR-21, miR-155, miR-182, miR-373, miR-199a, miR-205, and miR-375 and found that there is a significant difference (*p* < 0.05) in expression fold-changes between stage II and III patients regarding miR-21 (3.1 ± 1.1 and 6.1 ± 0.5, respectively), miR-155 (1.9 ± 0.8, and 4.2 ± 0.8, respectively) and miR-182 (3.5 ± 0.7 and 5.4 ± 1.0, respectively). Nonetheless, the expression of the aforementioned miRNAs did not show significant differences between patients with luminal A and luminal B subtypes. Moreover, among patients with regional lymph node involvement, miR-182 was remarkably higher (fold-change 2.9 ± 1 for N0 vs. 6.5 ± 1.9 for N1-3). Furthermore, the expression of the miRNAs in patients with different sensitivity to neoadjuvant polychemotherapy (NPCT), miR-205 and miR-375 proved to be better biomarkers for the assessment of tumor sensitivity in luminal A subtype; for instance, the increased levels of miR-205 (fold-change > 4.0) and reduced levels of miR-375 (fold-change < 0.3) exhibited higher sensitivity to fluorouracil + doxorubicin + cyclophosphamide and doxorubicin + cyclophosphamide regimens. Meanwhile, for the luminal B subtype, sensitive tumors were characterized by the expression fold-changes of miR-21 and miR-205 of less than 2.0 and greater than 3.0, respectively. The authors investigated the 3-year disease-free survival in luminal A patients and discovered that those with miR-182 expression fold-changes greater than 5.5 and miR-199a levels less than 0.2 had lower survival rates than those with miR-182 fold-change below 5.1 and miR-199a levels above 0.34 (*p* = 0.005). Moreover, the levels of miR-155 and miR-375 in luminal B subtype patients were able to predict disease relapse (*p* = 0.005). These miRNAs target genes are involved in cancer pathology. For instance, miR-21 targets the tumor suppressor LZTFL1 gene and promotes BC EMT through β-catenin [104], while miR-182 promotes cell proliferation and migration in TNBC via FOXF2 [105]. Huang et al. [106] reported that GPER inhibits TNBC cell proliferation, migration, and angiogenesis through the modulation of the miR-199a-3p/CD151 axis, which simultaneously inactivates the Hippo signaling pathway. Moreover, miR-205 has been reported to inhibit proliferation and promote apoptosis in TNBC cells by directly targeting CLDN11 [107]. Finally, Fu et al. [108] found that miR-375 directly targets HOXB3, which is associated with the formation of BCSC phenotypes, tamoxifen resistance, and the promotion of EMT in HER+ BC. Considering this, miR-21, miR-155, miR-182, miR-199a, miR-205, and miR-375 can potentially be used in clinical practice as predictive markers for drug-based therapy regimens, as well as markers for relapse-free survival [64].

McGuire et al. [60] analyzed a panel of five miRNAs (let-7a, miR-21, miR-145, miR-155, and miR-195) extracted from whole blood in response to NAT. Their differential expression levels were found to be related to several clinicopathological parameters. For example, miR-195 levels were higher in grade 2 BC than in grade 3 BC (*p* = 0.016), and increased titers of this miRNA were associated with ER+BC (*p* = 0.014). Previously, it has been reported that miR-195 can downregulate Mitofusin-2 (MFN2) in BC cell lines, affecting mitochondrial morphology and function, and inducing defects in mitochondrial respiration, altering the intrinsic apoptosis pathway [109]. In addition, lower levels of miR-195 and miR-21 were observed in responders compared to non-responders (*p* = 0.036 and 0.017, respectively). No significant differences were observed in circulating let-7a, miR-145, and miR-155 after NAT. These results suggest that miR-21 and miR-195 are potentially good predictors of the response to NAT in BC patients [60].

In their 2020 study, Di Cosimo et al. [55] analyzed plasma samples obtained from HER2+BC patients undergoing trastuzumab-based NAT as part of the NeoALTTO trial. The goal was to observe changes in miRNA levels during the first two weeks of treatment and to investigate any potential connection between early miRNA dynamics and patient response to NAT and clinical outcomes. Notably, miR-148a-3p and miR-374a-5p were elevated in the plasma samples of patients who achieved pCR (*p* = 0.008 and 0.048, respectively). Interestingly, the initial levels of these miRNAs did not correlate with treatment response, but their dynamics provided insights into the ultimate clinical response, indicating that these miRNAs are implicated in the mechanism underlying trastuzumab sensitivity and could be utilized as early indicators of treatment response. Remarkably, miR-148a-3p, a crucial tumor suppressor factor, was significantly downregulated in BC, which is closely linked to tumor grade and nodal involvement [110]. The role of miR-148a-3p in suppressing tumor growth has been demonstrated through its specific targeting of PKM2, a canonical protein kinase critical for maintaining the malignant phenotype, cell cycle progression, and tumor growth in cancer cells [111]. Additionally, studies have shown that the expression of miR148a-3p is directly regulated by the transcription factor EGR1 [111]. Interestingly, EGR1 is induced by treatment with anti-HER2 targeted therapies such as trastuzumab, and its expression is associated with favorable outcomes in HER2+ cancer models [112]. However, the precise role of miR-374a in BC remains elusive. In vitro studies have confirmed that miR-374a-5p promotes cancer progression, at least in TNBC, by targeting ARRB1 [91]. Given that the dynamics of plasma miRNAs might serve as a potential marker of chemosensitivity, its clinical relevance and the specific molecular pathways involved should be further validated in a larger series of patients with BC.

In a similar study, Zhang et al. [113] analyzed the expression of serum miR-222-3p to determine its contribution to the early prediction of therapeutic response, clinical outcomes, and adverse events in HER2+ BC patients receiving NAT. Serum samples were obtained before NAT from women with locally advanced invasive BC. During treatment, paclitaxel, cisplatin, and trastuzumab were administered to all patients. Results revealed that HER2+ BC patients with low miR-222-3p levels were more likely to achieve pCR, while higher levels were related to a minor pCR rate (*p* = 0.043). Interestingly, PTEN/Akt/FOXO1 signaling has been associated with adriamycin (ADR) resistance, and the overexpression of miR-222-3p is highly related to poor overall survival in BC [114]. Hence, miR-222-3p could serve as a potential noninvasive biomarker to predict pCR, survival, and trastuzumab-induced cardiotoxicity in HER2+ BC patients receiving NAT. However, large-scale studies are needed to elucidate exact roles in each condition [113].

Ritter et al. [65] investigated the biomarker potential of 12 miRNAs (let-7a, let-7e, miR-7, miR-9, miR-15a, miR-17, miR-18a, miR-19b, miR-21, miR-30b, miR-222, and miR-320c) in NAT for TNBC. For this purpose, they analyzed the expression of miRNAs in the serum of eight patients with TNBC. They noticed that in the serum of patients, let-7a (*p* = 0.008), let-7e (*p* = 0.005), and miR-21 (*p* = 0.039) were significantly upregulated, whereas miR-15a (*p* = 0.008), miR-17 (*p* = 0.023), miR-18a (*p* = 0.015), miR-19b (*p* = 0.002), and miR-30b (*p* < 0.001) were downregulated compared with the control group. Nonetheless, due to the small sample size, authors were not able to find a pattern for pCR nor for responder status. However, they found a pattern in the serum of patients achieving a clinical complete response (cCR) during NAT; the patients displayed an altered regulation trend immediately prior to the third cycle of therapy (t1) than those who did not achieve cCR. A significant downregulation was observed in miR-17 (*p* = 0.029), miR-19b (*p* = 0.03), and miR-30 (*p* = 0.011) levels between samples at the time prior to NAT (T0) and T1 when cCR was achieved. Zhong et al. [115] reported that in BC, doxycycline exerts its inhibitory effects via a miR-17-dependent pathway, where doxycycline binds to PAR1, inactivating NF-κB and downregulating miR-17, resulting in an upregulation of E-cadherin and an inhibition of EMT progression, decreasing BC migration. Furthermore, miR-19b regulates BC metastasis by directly targeting MYLIP; likewise, it modulates other cell adhesion molecules, such as E-Cadherin, ICAM-1, and Integrin β1 [116]. Finally, EZH2, a key epigenetic regulator, modulates miR-29b expression, which targets LOXL4 and contributes to cell proliferation, migration, metastasis, and immune microenvironment remodeling in BC [117]. While the small sample size limited the authors’ ability to draw definitive conclusions about the eligibility of the panel for predicting therapy response in vivo, their study highlights the potential of these miRNAs in liquid biopsies for evaluating the response of patients with TNBC to NAT. Further studies in larger cohorts are necessary to confirm the potential of this panel and its utility in clinical practice.

Other BC-related circulating miRNAs have attracted interest because of their potential use as predictors of NAT response in luminal B BC patients. Zhang et al. [14] performed a study using serum samples from several BC cohorts: grade I/II, grade III, HER2+, and HER2− to investigate the predictive effect of miRNAs on NAT response. Results demonstrated that miR-210 levels were significantly increased due to NAT response in all cohorts (*p* < 0.05). miR-210 is known to be involved in mechanisms related to anthracycline and taxane resistance, and its upregulation has been linked to poor prognosis in BC due to the targeting of the protein inhibitor of activated signal transducer and the activator of transcription 4 (PIAS4), which affects the sensitivity of BC chemotherapy and decreases apoptosis [118,119]. In addition, miR-222 was downregulated in the HER2+ group and patients without pCR (*p* = 0.039). A potential target of miR-222 is ANXA3, whose expression has been associated with poor prognosis and tumor growth [120]. A decreased expression of ANXA3 due to the overexpression of miR-222 leads to lower BC cell aggressiveness by declining cell proliferation, invasion, and migration [120]. Additionally, miR-375 was upregulated in HER2− patients and patients who did not achieve pCR (*p* = 0.006). A known target of miR-375 is HOXA5, whose underexpression promotes the evasion of apoptosis and stimulates cell proliferation and migration in BC [121]. The substantial downregulation of let-7g was also observed in patients without pCR. Owing to the negative regulation of FOXC2, let-7g plays a crucial role in BC cell migration [122]. These findings suggest that circulating miRNAs might indicate NAT response and prognosis in luminal BC [14].

Baldasici et al. [57] conducted a study to assess the use of plasma-circulating miRNAs as predictors of Miller–Payne’s (MP) pathological response to NAT in BC patients. Results revealed that the reduced expression of miR-21-5p (*p* = 0.0004), miR-221-3p (*p* = 0.0008), miR-146a-5p (*p* = 0.017), and miR-26a-5p (*p* = 0.045) was significantly associated with a positive response to NAT, as determined by the MP pathological response. MiR-21-5p, a well-known oncogenic miRNA, has been extensively linked to chemoresistance in BC by targeting tumor suppressors such as PTEN and programmed cell death 4 (PDCD4) [123]. MiR-221 is also known to inhibit critical tumor suppressor pathways, including PTEN/Akt/mTOR signaling, thereby promoting the chemoresistance of BC cells [124]. Moreover, miR-146a-5p was shown to be downregulated in TNBC and can repress cell proliferation, migration, and EMT by targeting the transcription factor SOX5, which is crucially involved in the determination of cell fate [125]. Finally, miR-26a-5p has been identified as a trastuzumab-inducible miRNA that plays a vital role in resistance to trastuzumab therapy by directly targeting the CCNE2 transcript, a key cell cycle regulator. Through CCNE2, miR-26a-5p regulates downstream biological pathways involved in chemoresistance, such as APAF1-mediated apoptosis and PI3K/Akt signaling [126]. These miRNAs are promising biomarkers for predicting treatment responses in patients with BC.

In the study by Todorova et al. [63], women with ER+/PR+/Her2- or TNBC, stage I to III BC were enrolled, and all patients were histologically confirmed to have early stage invasive ductal carcinoma (IDC). Patients were treated with a combination of adriamycin and cyclophosphamide, and plasma samples were collected before and 14 days after the first cycle of NAT. A total of eight miRNAs found in the blood were able to indicate the response to NAT in patients diagnosed with early stage breast cancer. Notably, there were no notable variations observed between the two groups of patients (those who achieved pCR and those who did not) in terms of age, race, and the grade of breast cancer. The analysis of differentially expressed miRNAs at the baseline revealed an increase in miR-30b (*p* < 0.0001) and a decrease in miR-423 (*p* = 0.0005) and miR-328 (*p* = 0.0019) in patients who achieved pCR compared to those who did not. Further, miR-127 exhibited a significant upregulation (*p* < 0.0001) in patients with TNBC who achieved pCR compared to those who did not. Following the first NAT dose, patients with pCR showed a significant decrease in miR-141 levels (*p* < 0.0001) in their plasma compared to the baseline. On the other hand, the non-pCR group evidenced an upregulation of miR-34a and miR-183, as well as a downregulation of miR-182 compared to the baseline (*p* < 0.0001 for all three miRNAs). Interestingly, these miRNAs have been previously associated with BC carcinogenesis. For example, miR-30b-5p reduces cell apoptosis and promotes the EMT process, which is consistent with the effects of ASPP2 silencing in breast tumor samples [127]. Similarly, miR-328-3p has been shown to target and regulate COL1A1, associated with the elastic parameters of breast lesions [128,129], and miR-423-5p enhanced the NF-κB signaling activation pathway by directly targeting TNIP2, the vital negative regulator of the NF-κB pathway, finely modulating BC invasiveness [130]. Interestingly, the oncogenes CERK, NANOS1, FOXO6, SOX11, SOX12, FASN, SUSD2, and BZRAP1 are targets of miR-127-3p; therefore, this miRNA is considered a tumor suppressor [131,132]. MiR-141-3p targets CDK8 and restores sensitivity to trastuzumab by repressing this protein [133]. Moreover, miR-34a-5p inhibits cell proliferation and migration and promotes apoptosis by targeting B7-H1 [94], whereas PPP2CA was shown to be a target gene of miR-183-5p, contributing to tumor progression in BC models [134]. MiR-182-5p aggravates BC by downregulating CMTM7 and activating EGFR/AKT signaling pathway. Overall, significantly dysregulated plasma exosomal miRNAs before, during, and after NAT could be used as minimally invasive pCR predictors in BC. Additionally, the four miRNA signatures found are substantially associated with NAT therapeutic response. Nonetheless, a larger cohort of patients is still needed to confirm these results [63].

Davey et al. [66] conducted a study to determine the impact of miRNAs in predicting NAT response. For this purpose, blood samples were collected from 120 female patients who underwent NAT for BC at two independent time points: before NAT treatment (T1) and halfway during NAT (T2). They evaluated the expression patterns of a panel of five miRNAs, namely let-7a, miR-21, miR-145, miR-155, and miR-195 and compared the two time points. Authors found that upregulation of let-7a was useful in identifying responders to NAT in luminal/HER2+ disease (L/HER2) (*p* = 0.049); likewise, the downregulation of miR-21 identified responders in L/HER2 (*p* = 0.058). For HER2+ (ER-/PgR-/HER2+) patients, the underexpression of miR-145 (*p* = 0.027) and let-7a identified NAT responders, whereas the overexpression of miR-21 identified responders in TNBC. Moreover, decreased let-7a (*p* = 0.037) and increased miR-145 levels predict pCR in patients with luminal BC. Likewise, in HER2+ patients, downregulated miR-145 (*p* = 0.027) and let-7a predicted pCR. Finally, diminished miR-21 levels were associated with the prediction of pCR. These results confirm the observations of Liu et al. [135] stating that let-7a acts as a tumor suppressor to decrease proliferation by directly targeting the USP32 gene in BC. Thus, let-7a, miR-21, and miR-145 could be used as biomarkers to determine the response to NAT and/or predict pCR, while miR-195 and miR-155 expression failed to predict sensitivity to NAT. Hence, these preliminary results require further validation before their actual use in clinical practice [66].

Recently, Sukhija et al. [59] analyzed the effect of chemotherapy on miR-21 levels in metastatic BC. The responses to candidates of NAT, including adriamycin, cyclophosphamide, and taxane, were evaluated. MiR-21 was obtained from whole blood, and the subjects underwent three NAT cycles. After the treatment, a 5.65-fold overexpression of miR-21 was observed; however, there was also no significant correlation with clinical outcome [59].

## 3. Discussion

Circulating miRNAs are useful biomarkers for monitoring treatment response in BC since they can provide real-time information about the tumor microenvironment and systemic effects of therapy. However, the burgeoning interest in using circulating miRNAs as blood-based biomarkers requires a meticulous examination of the potential impact of various preanalytical and analytical parameters on their measurement accuracy. Particularly, preanalytical factors, including sample handling and storage conditions prior to processing, play a critical role in ensuring the reliability and reproducibility of circulating miRNA quantification. Moreover, a significant technical challenge in studying miRNA expression lies in the reliable and efficient extraction of these molecules from blood samples due to their small size and their association with lipids and proteins. Thus, oncologists should work in close collaboration with research scientists and laboratory staff to obtain samples from patients and process them following validated protocols. The use of circulating miRNAs as biomarkers for monitoring treatment response against NAT can identify patients who are not responding to therapy early on, allowing for a switch to alternative therapies. However, much work needs to be done to optimize diagnostic procedures, reduce preclinical variability, and standardize the methods for assessing circulating miRNAs.

Large-scale, prospective, multicenter studies are needed for determining the predictive and therapy-monitoring utility of circulating miRNAs. Once standardized methods are developed, these biomarkers could serve as a novel additional diagnostic, prognostic, and disease-monitoring factor in clinical practice. From a technical standpoint, miRNAs possess optimal properties that make them easily accessible as prognostic biomarkers. These small RNA molecules are highly stable and exhibit a long half-life in biological samples. Furthermore, their analysis does not require specialized handling and can be performed using standard techniques already employed in clinical laboratories, such as quantitative PCR. This makes miRNA quantification relatively cost-effective, sensitive, and specific. Nonetheless, there are also some limitations associated with the use of miRNAs in clinical practice. The need for further standardization and validation of miRNA detection methods poses a challenge to the reproducibility of results across different laboratories and clinical settings. Furthermore, miRNA expression can be influenced by various biological and environmental factors, such as age, gender, ethnicity, and lifestyle habits, challenging their clinical utility. Despite these limitations, the potential of miRNAs as diagnostic and therapeutic tools in BC NAT is enormous. With ongoing research and development, miRNAs could serve as reliable biomarkers for the early detection and prognosis of BC and for monitoring treatment response as well as identifying drug resistance mechanisms.

## 4. Conclusions

Herein, we provided an overview of recent research on the dynamics of circulating miRNAs, their role in evaluating the response to NAT in breast cancer, and their clinical potential. Overall, through serial assays to assess circulating miRNA profiles during NAT, most of the research demonstrated significant fluctuation patterns in plasma/serum, suggesting that miRNA dynamics may serve as a novel response indicator to guide the personalized delivery of NAT. In conclusion, the use of circulating miRNAs as a diagnostic and therapeutic tool in BC NAT has shown great promise in improving patient outcomes. Although some limitations and challenges are still associated with their use, ongoing research and development will likely continue to refine and validate their clinical utility. The future of BC patient management may be revolutionized by integrating circulating miRNAs into clinical practice, but careful validation and standardization are required to ensure their clinical efficacy. Overall, miRNA-based approaches have the potential to improve the diagnosis, treatment, and management of BC and increase patient survival and quality of life.

## 5. Future Directions

Emerging omics technologies are revolutionizing medical decision-making, and among transcriptomic tools, miRNAs have emerged as being particularly promising. The dysregulation of miRNAs disrupts intricate regulatory networks and is a prevalent characteristic of various diseases, including BC [136]. This suggests that analyzing miRNA profiles could lead to innovative approaches for diagnosis, prognosis, and personalized therapies. Consequently, in recent years, numerous research areas have concentrated on disease management by restoring the balance of dysregulated processes through the development of miRNA-based treatments [137]. Two strategies can be employed to harness this potential: the depletion of oncogenic miRNAs or the enhancement of tumor-suppressive miRNAs [138,139]. To deplete oncogenic miRNAs, an oligomer that complements the target miRNA, known as an antagomir, can be delivered. Antagomirs bind to mature miRNAs, hindering their function and leading to degradation. Conversely, enriching tumor-suppressive miRNAs entails the delivery of miRNA mimics, which are double-stranded RNA sequences with the same sequence as the targeted miRNA. Over the past decade, efforts have been made to transform miRNA-based therapeutics into cancer treatments by minimizing off-target effects [140]. However, several significant challenges hamper the successful clinical application of miRNA-based drugs [141]. These challenges include designing targeting sequences to enhance on-target specificity, modifying oligonucleotides to improve stability and cellular uptake, determining the appropriate dosing for improved efficacy and potency, achieving cell/tissue-specific delivery, avoiding an immune response, and utilizing biocompatible and biodegradable carrier materials [142]. The number of clinical trials involving miRNA-based therapeutics is rapidly increasing. Despite these promising advancements, miRNA-based drugs still need to complete phase II clinical trials and receive approval from the FDA before they can be used clinically [139]. Undoubtedly, miRNAs have immense potential in providing avenues for diagnosis, prognosis, and therapy. As the field progresses, we anticipate that miRNA-based therapeutics could pave the way for novel categories of medications targeting a wide range of diseases including BC.

## Figures and Tables

**Figure 1 ijms-24-09984-f001:**
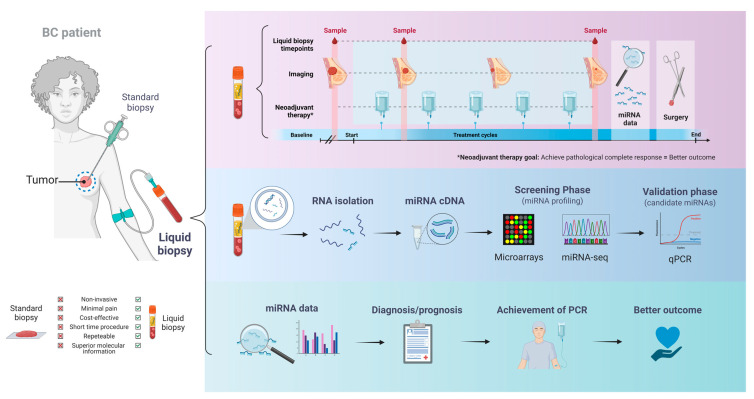
Schematic representation of circulating miRNA utility as biomarkers for breast cancer diagnosis/prognosis in response to neoadjuvant therapy.

**Table 1 ijms-24-09984-t001:** Dysregulated circulating miRNAs in response to NAT in BC patients.

Associated miRNAs	Target mRNA	Altered Biological Mechanism	Source	Reference
miR-222 ↑	TRPS1 Notch3	EMT	Plasma	[52]
miR-20a ↑	ZBTB4	Cell growth and invasion		
miR-451 ↓	MRP-1	Anthracycline resistance		
miR-34a ↓	Bcl-2 SIRT1	Apoptosis		
miR-155 ↓	ErbB2 HDAC2	Cell survival, growth, and chemosensitivity	Plasma	[53]
miR-301 ↓	ESR1	Cell growth and invasion		
miR-146b-5p ↓	ZEB1	EMT	Plasma	[54]
miR-199a-5p ↑	CDH1, ZEB1, TWIST			
miR-148a-3p ↑	PKM2	Malignant phenotype maintenance	Plasma	[55]
miR-374a-5p ↑	ARRB1	Cell survival, proliferation, and migration		[55,56]
miR-21-5p ↓	PTEN PDCD4	Chemoresistance	Plasma	[57]
miR-221-3p ↓	PTEN			
miR-146a-5p ↓	SOX5	EMT		
miR-26a-5p ↓	CCNE2	Chemoresistance		
miR-21 ↓	PTEN	Apoptosis and cell proliferation	Serum	[58]
miR-21 ↑	LZTFL1	Cell proliferation, invasion, and migration	Whole blood	[59]
miR-195 ↓	MFN2	Mitochondrial metabolism		[60]
miR-210 ↑	PIAS4	Apoptosis, anthracycline resistance	Serum	[14]
miR-222 ↓	ANXA3	Cell proliferation, invasion, and migration		
miR-375 ↑	HOXA5	Apoptosis, cell proliferation, invasion, and migration		
let-7g ↓	FOXC2	Cell migration		
miR-21-3p ↑	MAT2 STARD13 ZNF132	Cell proliferation, invasion, and migration	Serum	[61]
miR-145-5p ↑	SOX2	Cell proliferation		
miR-145 ↑	HBXIP			
miR-10b-5 ↑	BIRC5 E2F2 KIF2C FOXM1 MCM5			
miR-181a ↑	SOCS3 PIAS3 ATM	Cell proliferation, invasion, and migration	Whole blood	[62]
miR-181a-5p ↑	NDRG2			
miR-105 ↑	ZO-1			
miR-221 ↑	ERα			
miR-222 ↑	ERα			
miR-155 ↑	PTEN			
miR-30b-5p ↑	ASPP2	Apoptosis and EMT	Plasma	[63]
miR-328-3p ↑	COL1A1	Inflammation		
miR-423-5p ↑	TNIP2	Cell invasion		
miR-127-3p ↑	CERK NANOS1 FOXO6 SOX11 SOX12 FASN SUSD2 BZRAP1	Cell proliferation, invasion, and migration		
miR-141-3p ↑	CDK8	Chemoresistance		
miR-34a-5p ↑	B7-H1	Apoptosis, cell proliferation, invasion, and migration		
miR-183-5p ↑	PPP2CA	Cell proliferation, invasion, and migration		
miR-182-5p ↑	CMTM7	Cell proliferation, invasion, and migration		
miR-376c-3p ↓	RAB2A	Malignant phenotype maintenance Cell stemness	Plasma	[56]
miR-197-3p ↓	HIPK3	Apoptosis, cell proliferation, invasion, and migration		
miR-874-3p ↑	SOX2	Tumor growth		
miR-320c ↑	AQP1	Cell proliferation		
miR-100-5p ↑	PlK1	Cell proliferation, invasion, and migration		
miR-144-3p ↑	SOX2	Tumor growth Cell stemness		
miR-362-3p ↑	hERG	Apoptosis, cell proliferation, invasion, and migration		
miR-374a-5p ↑	ARRB1	Tumor progression		
miR-574-3p ↑	ADAM28	Tumor progression		
miR-140-5p ↑	Wnt1	Cell proliferation, invasion, and migration		
miR-145-5p ↑	SOX2	Tumor growth Cell stemness		
miR-328-3p ↓	Ki-67	Cell proliferation		
miR-34a-5p ↓	PD-L1 P53	Apoptosis, cell proliferation, invasion, and migration		
miR-98-5p ↑	IGF1	Cell proliferation, invasion, and migration		
miR-182 ↑	FOXF2	Cell proliferation and migration	Serum	[64]
miR-375 ↓	HOXB3	Malignant phenotype maintenance Cell stemness Chemoresistance		
miR-205 ↑	CLDN11	Apoptosis and cell proliferation		
miR-21 ↓	LZTFL1	EMT		
miR-199a ↓	CD151	Cell proliferation, invasion, and migration Angiogenesis		
miR-155 ↑	SOCS1 MMP16	Cell proliferation and migration		
miR-17 ↓	E-cadherin	EMT	Serum	[65]
miR-19b ↓	MYLIP E-cadherin ICAM-1 Inregrin β1	Cell adhesion		
miR-30 ↓	LOXL4	Cell proliferation, invasion, and migration		
let-7a ↑	USP32	Cell proliferation	Whole blood	[66]
miR-21 ↑	LZTFL1	Cell proliferation, invasion, and migration		
miR-145 ↓	SOX2	Tumor growth Cell stemness		

↑ Upregulated (overexpressed) miRNA; ↓ Downregulated (underexpressed) miRNA.

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
