# Peer review of "Exploring the Potential Role of Circulating microRNAs as Biomarkers for Predicting Clinical Response to Neoadjuvant Therapy in Breast Cancer"

_ijms, 2023, doi:10.3390/ijms24129984_

Round 1

Reviewer 1 Report

The present review article, ijms-2420649 entitled (Exploring the potential role of circulating microRNAs as biomarkers for predicting clinical response to neoadjuvant therapy in Breast Cancer)

The current review article contained an important topic work, good analysis and written in a excellent way. However, I have some minor comments as following:

-          - Please add list of all abbreviations at the end of the review.

-          - Please add (P-value) to the significant results to illustrate level of significance.

-      -Please use one reference style following journal guidelines for writing references inside text as there are two different styles in the review.

-          In page (3) line 107 reference [36] in not (W. Zhu and collaborators ???)- Please check

-          In page (15) this part is more likely to be a discussion and conclusion not as a discussion only.

-          In page (15) line 528 ***effects of therapy. While it is true**** I think these are two sentences , please check. ???  

 Minor editing of English language required

Reviewer 2 Report

This is a well-reviewed manuscript about circ miRNAs in BC samples.

A few suggestions,

1.     Main text is too long. Shorten it.

2.     Line 78. miRNA also binds to the coding region of mRNA.

Reviewer 3 Report

In this review the authors summarize recent data on circulating miRNAs in the neoadjuvant treatment of breast cancer. Identifying predictive biomarkers for this therapy is important as therapy-resistant cancers are frequently observed. So the topic is interesting for the community. Similar reviews are present in the recent literature, so this one does not provide too much additional information. Nevertheless, I am not opposing publishing another one in this journal.

The manuscript is generally well written and gives a well-organized overview on the topic.

Overall, the authors have identified about 17 original publications on this subject and listed these comprehensively in table 1. The results of the papers are well summarized and set into the clinical context.

There are a few typos that should be corrected; E.g. I think in table 1 “noch3” means notch3?

English is fine, only minor corrections required.

Reviewer 4 Report

The review article offers a comprehensive overview of the role of miRNAs in breast cancer, with a particular focus on their potential as predictive markers for response to neoadjuvant therapy (NAT). The paper successfully captures a wide range of studies from as early as Sassen et al. (2008) to the more recent findings by Davey et al. (2022) and Sukhija et al. (2023).

The depth and breadth of discussion on each of the identified miRNAs (miR-10b, miR-21, miR-125b, miR-145, miR-155, miR-195, and let-7a) is commendable. The authors effectively highlight the dualistic role that some miRNAs play in cancer pathology, acting either as tumor suppressors or oncogenes depending on their molecular context.

The piece is particularly strong in its presentation of the differing roles that these miRNAs can play in the various subtypes of breast cancer and in their potential to serve as predictive markers for NAT response and pathologic complete response (pCR).

The authors critically acknowledge the limitations of current research, such as the limited scope of miRNA panels and the need for inclusion of all molecular subtypes of breast cancer. The recognition of the need for additional research for clinical validation is crucial and appreciated.

However, there are some areas where the review could be strengthened:

1 ) A more detailed exploration of the potential mechanisms underlying miRNA dysregulation in breast cancer would help provide context for the changes seen in their expression levels.

2 ) The paper could benefit from a broader discussion on the implications of these miRNAs in the overall clinical management of breast cancer. While some points are mentioned, this could be further expanded to discuss future therapeutic strategies and the potential challenges in their implementation.

3) The methodology section, especially relating to miRNA expression assessment from blood samples, needs more attention. Understanding the nuances of these techniques could provide a clearer picture of the robustness of the results.

4) Additionally, while the paper identifies several promising miRNAs as potential predictive biomarkers, it would be helpful to discuss the potential strategies for using these miRNAs in clinical settings and the hurdles that must be overcome to achieve this.

5) Also I think its worth dedicating a chapter to the known mechanisms of action of NAT in the beginning.

With the above suggestions, this paper could serve as a very comprehensive guide for both researchers and clinicians in the field.
